# Circulating Cell-Free DNA as a Prognostic Biomarker in Resectable Ampullary Cancer

**DOI:** 10.3390/cancers13102313

**Published:** 2021-05-12

**Authors:** Bor-Uei Shyr, Bor-Shiuan Shyr, Shih-Chin Chen, Shih-Ching Chang, Yi-Ming Shyr, Shin-E Wang

**Affiliations:** 1Department of Surgery, National Yang Ming Chiao Tung University, Taipei 11221, Taiwan or bwshyr@vghtpe.gov.tw (B.-U.S.); or bsshyr@vghtpe.gov.tw (B.-S.S.); or scchen19@vghtpe.gov.tw (S.-C.C.); changsc@vghtpe.gov.tw (S.-C.C.); 2Department of Surgery, Taipei Veterans General Hospital, Taipei 11217, Taiwan

**Keywords:** ampullary, cancer, cfDNA, pancreaticoduodenectomy

## Abstract

**Simple Summary:**

Circulating cell-free DNA (cfDNA) in patients with ampullary cancer was measured to clarify the correlation between cfDNA and clinicopathological factors and the impact of cfDNA on survival outcomes. The level of cfDNA was significantly higher in patients with lymph node involvement, lymphovascular invasion, abnormal serum carcinoembryonic antigen level, and stage II and III cancer. The 1- and 5-year survival rates were 92.0% and 66.5%, respectively, for patients with low cfDNA levels ≤ 6687 copies/mL as compared with 84.0% and 49.9%, respectively, for patients with high cfDNA levels > 6687 copies/mL (*p* < 0.001). After multivariate analysis, only the cfDNA level and cancer stage were independent factors in determining the prognosis of the ampullary cancer. The cfDNA level could act as a surrogate marker of both disease extent and biological aggressiveness of ampullary cancer. Moreover, cfDNA plays a significant role in determining the prognosis of resectable ampullary cancer.

**Abstract:**

Circulating cell-free DNA (cfDNA) in ampullary cancer patients was measured to clarify the correlation between cfDNA and clinicopathological factors and the impact of cfDNA on survival outcomes. Patients with ampullary cancer undergoing pancreaticoduodenectomy were included. Correlations between cfDNA and clinicopathological and prognostic factors were determined. The cfDNA levels in patients ranged from 1282 to 21,674 copies/mL, with a median of 6687 copies/mL. The cfDNA level was significantly higher in patients with lymph node involvement, lymphovascular invasion, abnormal serum carcinoembryonic antigen (CEA) level, and stage II and III cancer. Poor prognostic factors for ampullary cancer included high cfDNA > 6687 copies/mL, lymph node involvement, abnormal serum CEA > 5 ng/mL, and advanced stage II and III cancer. The 1- and 5-year survival rates were 92.0% and 66.5%, respectively, for patients with low cfDNA < 6687 copies/mL and 84.0% and 49.9%, respectively, for patients with high cfDNA > 6687 copies/mL (*p* < 0.001). After multivariate analysis, only the cfDNA level and stage were independent prognostic factors of ampullary cancer. Thus, the cfDNA level could act as a surrogate marker of both disease extent and biological aggressiveness of ampullary cancer. Moreover, cfDNA plays a significant role in the prognosis of resectable ampullary cancer.

## 1. Introduction

Circulating cell-free DNA (cfDNA) in plasma has attracted great interest in cancer research due to its noninvasive manner of use in cancer diagnosis and monitoring, or the so-called “liquid biopsy” [1,2]. Therefore, understanding its molecular characteristics would provide valuable information that might lead to the early detection and accurate prediction of oncological outcomes. Some studies have shown that patients with malignancy are associated with higher levels of cfDNA than healthy individuals [1,3]. There is an increase in cell turnover during tumorigenesis, resulting in more necrotic and apoptotic cells, which could be released into the bloodstream and lead to an accumulation of cfDNA [4]. This theory might explain why cancer patients tend to have higher cfDNA levels than normal healthy patients [5]. Therefore, based on the concept of “liquid biopsy”, cfDNA has recently been considered as a promising prognostic biomarker in patients with various types of cancer [1,6,7,8,9,10,11,12].

Ampullary cancer is relatively rare as compared with pancreatic head cancer, accounting for only 0.2% of gastrointestinal cancers and approximately 16% to 28% of all periampullary cancers [13]. Patients with ampullary cancer continue to experience poor outcomes. Therefore, it is crucial to identify potential biomarkers for early detection, novel therapeutic strategies, and determination of the prognosis in patients with ampullary cancer. Nevertheless, there is relatively limited data regarding diagnosis and prognosis for ampullary cancer, largely because of the rarity of the disease and the paucity of related research. To our knowledge, there is no report on the application of cfDNA in ampullary cancer, although studies of cfDNA in other malignancies are extensive.

This study measured plasma cfDNA levels in patients with resectable ampullary cancer. The correlations between cfDNA and clinicopathological factors were also clarified. The prognostic factors for ampullary cancer were determined, and the impact of cfDNA on survival outcomes was evaluated using univariate and multivariate analyses.

## 2. Results

The cfDNA level was measured in 100 patients with ampullary cancer undergoing pancreaticoduodenectomy. The cfDNA levels in all patients ranged from 1282 to 21,674 copies/mL with a median of 6687 copies/mL and a mean of 7455 ± 4027 copies/mL. The median of 6687 copies/mL was used as the cut-off for analysis. The levels of cfDNA in 95 normal, healthy controls ranged from 0 to 4157 copies/mL with a median of 168 copies/mL and a mean of 613 ± 888 copies/mL at our institute (Figure 1). The ROC curve (receiver operating characteristic curve) is shown in Figure 2. The level of cfDNA had no correlations with age, sex, tumor size, tumor cell differentiation, perineural invasion, or serum level of CA 19-9. Moreover, cfDNA levels were significantly higher in patients with lymph node involvement (median: 7569 vs. 5492 copies/mL, *p* < 0.001), lymphovascular invasion (median: 7569 vs. 5768 copies/mL, *p* = 0.0029), abnormal serum CEA level (median: 11,627 vs. 6338 copies/mL, *p* < 0.001), and stage II and III cancer (median: 6893 vs. 5697 copies/mL, *p* = 0.0028) (Table 1).

Prognostic factors for ampullary cancer after pancreaticoduodenectomy are listed in Table 2. By univariate analysis, the poor prognostic factors included a high cfDNA level > 6687 copies/mL (cut-off value based on the median cfDNA level for total patients), positive lymph node involvement, abnormal serum CEA level > 5 ng/mL, and advanced stage II and III cancer.

The 1- and 5-year survival rates were 92.0% and 66.5%, respectively, for patients with a low cfDNA level ≤ 6687 copies/mL, as compared with 84.0% and 49.9%, respectively, for patients with a high cfDNA level > 6687 copies/mL, *p* < 0.001 (Figure 3).

After multivariate analysis by the Cox proportional hazards regression model, only the cfDNA level and cancer stage were determined as independent factors to ascertain the prognosis of patients with ampullary cancer (Table 3).

## 3. Discussion

The most reliable prognostic markers for ampullary cancer may be lymph node involvement, grading, serum CEA level, and cancer staging [14,15]. This study demonstrated that not only these traditional factors but also the biomarker of cfDNA level, had prognostic value for patients with ampullary cancer. Moreover, cfDNA refers to fragmented DNA, consisting of 70–200 base pair (bp) fragments, and found in the noncellular component of the blood, as first reported by Mandel et al. in 1948 [16]. It is thought that cfDNA is released into the bloodstream through apoptosis or necrosis of cells, and it is usually detected as double-stranded fragments of approximately 150 to 200 bp in length [16]. Although it might be actively released from normal cells as a part of metabolism, 4–40-fold greater levels could be detected in patients with malignancy [5,7,8,10]. Recently, analysis of cfDNA has rapidly emerged as a type of liquid biopsy, providing a less invasive approach to diagnose cancers, monitor chemotherapy-resistant mutations, and overcome tumor heterogeneity [16]. Apart from cfDNA analysis, liquid biopsy can also be carried out by measuring the circulating tumor DNA (ctDNA), circulating tumor cells (CTCs), and other nucleic acids such as microRNA (which is less stable than DNA) in blood [16]. Ideally, CTCs or ctDNA would be more specific as cancer biomarkers. However, CTCs are usually present in very low concentrations of less than 10 CTCs/mL of blood, even in patients with metastatic disease [16,17]. Therefore, this low concentration would result in low sensitivity for the detection of cancers, thus limiting its clinical application [18,19]. Ideally, ctDNA could be discriminated from normal cfDNA by detecting tumor-specific somatic mutations that exist only in the genomes of cancer cells and not in normal cells. Although the fraction of ctDNA tends to parallel the tumor burden within a cancer patient, detection of ctDNA is also challenging because the fraction of ctDNA within the total cfDNA in cancer patients could vary greatly from less than 0.1% to more than 90% [16]. Therefore, cfDNA could be a practical potential biomarker in the field of liquid biopsy for ampullary cancer. The applicability of cfDNA in other tumor types, such as pancreatic cancer and stomach cancer, has been reported [1,3,6,9].

The results revealed that the cfDNA level was significantly higher in patients with positive lymph node involvement or stage II and III cancer than in those with negative involvement or stage I cancer. In other words, the cfDNA level tended to increase with advanced ampullary cancer. These associations might suggest that cfDNA levels could be a reflection of the tumor burden in patients with ampullary cancer. Moreover, higher levels of cfDNA were also noted in patients with positive lymphovascular invasion and abnormal serum CEA levels. These findings imply that the cfDNA level could be a biomarker of the biological behavior of ampullary cancer. After multivariate analysis, only the cfDNA level and cancer stage remained as independent prognostic factors. Therefore, the cfDNA level could act as a surrogate marker of both the extent of disease and biological aggressiveness of ampullary cancer.

There are few overlaps of cfDNA levels between healthy volunteers and cancer patients. There are some limitations of this study. The PCRs for the cfDNA samples of healthy volunteers and cancer patients were not performed simultaneously because it was impossible to collect the blood samples of healthy volunteers and cancer patients at the same time. Moreover, assays of cfDNA are sensitive to genomic DNA contamination derived from lysed cells in poorly manipulated samples, cfDNA degradation, and the presence of enzymatic inhibitors. Hence, it has been emphasized the need to standardize collection, handling, and preservation methods as well as the importance to perform consistent quality controls on isolated cfDNA. We did carefully follow rules to perform the assays of cfDNA of healthy volunteers and cancer patients, but technical error might still be inevitable since the blood samples of healthy volunteers and cancer patients were not collected at the same time and the PCRs for cfDNA samples for these two groups were not performed simultaneously.

## 4. Materials and Methods

Patients with ampullary cancer undergoing resection with pancreaticoduodenectomy from January of 2005 to October of 2018 were recruited for study. Data of these patients were prospectively collected and kept in a computer database. This study was exclusively conducted for the pathological diagnosis of adenocarcinoma in the ampulla of Vater. Pancreaticoduodenectomy without extensive retroperitoneal lymph node dissection was performed in all patients. Lymph node dissection was carried out along the superior mesenteric vein, including the tissues around the common hepatic artery, head of the pancreas, and hepatoduodenal ligament. All surgical procedures were performed by the same team led by YM Shyr using a technique that was previously described in detail [20,21]. The study was approved by the Institutional Review Board of the Taipei Veterans General Hospital (IRB-TPEVGH NO.: 2018-11-004BC). Informed consent was waived in this retrospective cohort study with anonymization of the data.

Correlations of the cfDNA level with various demographics and prognostic factors were evaluated, including age, gender, tumor size, lymph node involvement, tumor cell differentiation, lymphovascular invasion, perineural invasion, carbohydrate antigen 19-9 (CA 19-9), carcinoembryonic antigen (CEA), and tumor stage. Prognostic factors for ampullary cancer were clarified by univariate analysis, whereas independent prognostic factors were further determined by multivariate analysis with a Cox proportional hazards regression model. The method for measuring CA 19-9 was radioimmunoassay and CEA was based on chemiluminescence in serum during enzyme-linked immunosorbent assays (ELISA).

### 4.1. cfDNA Quantification

A blood sample was collected from each patient with written informed consent before surgery and stored in the Biobank of Taipei Veterans General Hospital. All blood samples were anonymous in our biobank. The cfDNA from each Biobank plasma sample was determined using a commercial QIAamp DNA Tissue Kit and MinElute Virus Kit (Qiagen, Valencia, CA, USA) according to the manufacturer’s instructions. The quality and quantity of plasma DNA were evaluated using a Nanodrop 1000 Spectrophotometer (Thermo Fisher Scientific, Waltham, MA, USA). A TaqMan quantitative polymerase chain reaction (qPCR) assay (Thermo Fisher Scientific) of the housekeeping gene cyclophilin, which was not known to be correlated with cancer, was used to quantify the cfDNA copy numbers in the plasma samples. qPCR was performed using TaKaRa Ex Master Mix (Takara Bio, Shiga, Japan) according to the manufacturer’s instructions. The sequence of cyclophilin primers were as below: forward ACATGGGTACTAAGCAACAAAATAAG and reverse CACAATTGGAACATCTTTGTTAAAC. The probe primer was Fam-TTGCAGACAAGGTCCCAAAGACAGCA-Tamra. Serially diluted standard DNA was used to generate a standard curve. The results were expressed as the threshold cycle (Ct), which was the cycle number at which the PCR product crossed the threshold of detection. To reduce the batch effect, we prepared a large volume tube of pre-mixed plasma samples (20 mL pooled from multiple samples) and prepared small aliquots of the pooled samples in standard tubes (1 mL) for storage at −80 °C. When performing cfDNA extraction and qPCR experiment, we used plasma samples from clinical individuals and the pre-mix standard tube. The cfDNA copy number in each patient was measured according to the Ct value and the standard curve from serially-diluted DNA (0.001, 0.01, 0.1, 1, 10, 100 ng). The results of standard tubes between different batches were used to calculate the batch-effect factor for adjusting the copies/mL value in the following analyses. The batch-effect factor was calculated based on the pre-mix standard plasma cfDNA level. Subsequently, the cfDNA copy number was normalized according to the plasma input volume and the batch-effect factor, and was expressed in copies/mL. The cfDNA levels of 95 healthy volunteers without any history of malignancy were measured at our institute, including 66 males and 29 females with a mean age 54.2 ± 15.5 years. These healthy volunteers had no subsequent diagnosis of malignancy for at least two years after blood sampling. There was no difference in the method of sample collection and DNA isolation between cfDNA samples of healthy volunteers and cancer patients. The samples were all from the same biobank and were all collected in the same type of collecting tube, but not during the same period of time. The PCRs for the cfDNA samples of healthy volunteers and cancer patients were not performed simultaneously because it was impossible to collect the blood samples of healthy volunteers and cancer patients at the same time, but using the same standard.

### 4.2. Statistical Analysis

Statistical analyses were carried out using Statistical Product and Service Solutions (SPSS) version 21.0 software (SPSS Inc., IBM, Armonk, NY, USA). All continuous data were presented as the median (range) and mean ± standard deviation (SD), and frequencies were presented when appropriate to the type of data. Mean values of continuous variables were compared with a two-tailed Student’s *t* test. Non-parametric statistical tests were used for variables that do not follow a normal distribution. Categorical variables were presented as numbers and percentages and were compared using Pearson’s χ^2^ test or Fisher’s exact test contingency tables. To determine the subset of factors that provided independent information on survival time, a Cox proportional hazards regression model was developed. For all analyses, *p* < 0.050 was considered statistically significant.

## 5. Conclusions

In conclusion, the level of cfDNA could be used as a surrogate marker to determine the extent and aggressiveness of ampullary cancer. Moreover, cfDNA plays a significant role in determining the prognosis of resectable ampullary cancer. Furthermore, by acting as a promising liquid biopsy, cfDNA could also be applied clinically to monitor not only treatment response but also tumor progression.

## Figures and Tables

**Figure 1 cancers-13-02313-f001:**
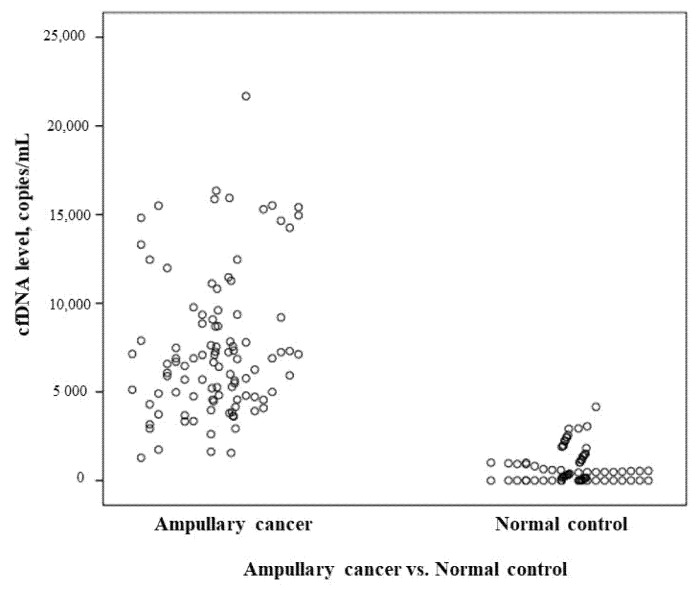
The scatterplot for ampullary cancer (study group) and healthy volunteers (normal control).

**Figure 2 cancers-13-02313-f002:**
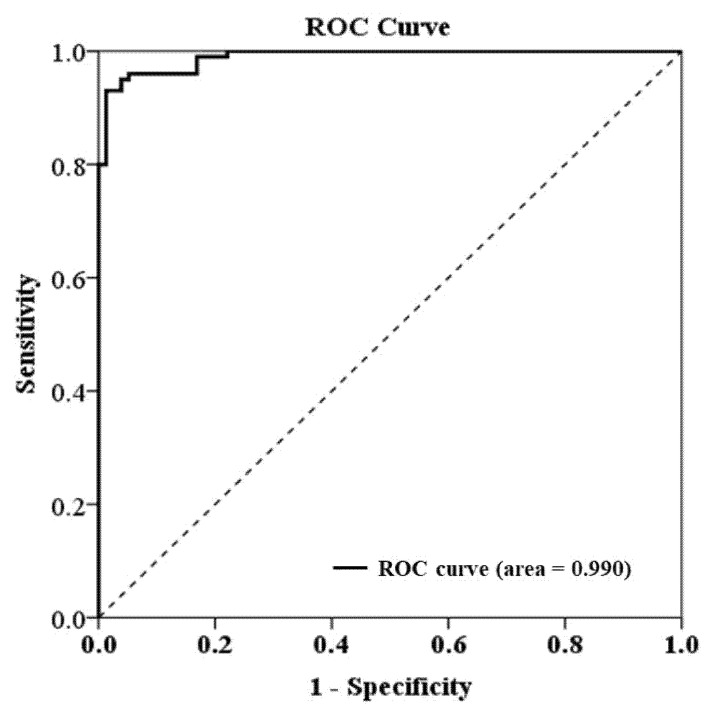
The ROC curve (receiver operating characteristic curve), ampullary cancer (study group) vs. healthy volunteers (normal control).

**Figure 3 cancers-13-02313-f003:**
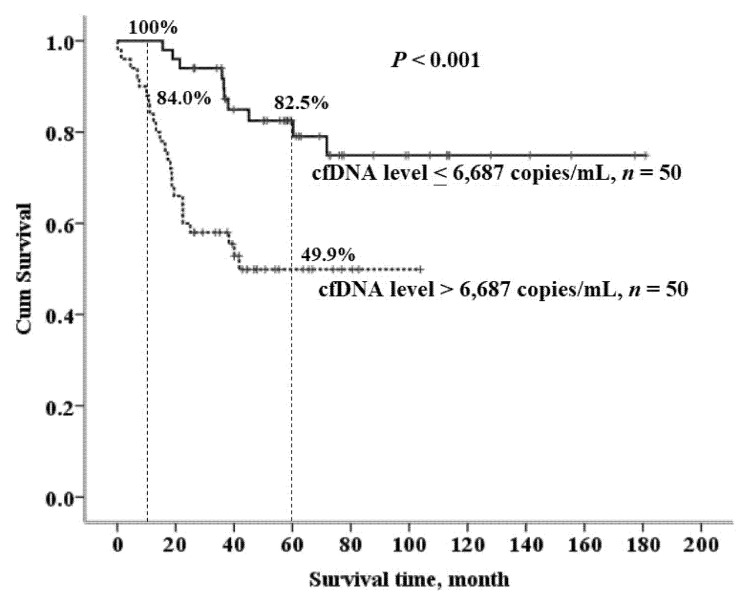
Survival outcomes for patients with pancreatic head adenocarcinoma after pancreaticoduodenectomy (PD) with high and low cfDNA levels.

**Table 1 cancers-13-02313-t001:** Circulating cell-free DNA in ampullary cancer patients undergoing pancreaticoduodenectomy.

Correlation Factor	Mean (SD)	Median	Range	*p*
Total, *n* = 100	7455 (4027)	6687	1292–21,674	
Age, y/o				0.386
≤65, *n* = 41	7033 (3665)	6981	1560–21,674	
>65, *n* = 59	7747 (4267)	6419	1292–16,345	
Sex				0.990
Male, *n* = 56	7459 (4105)	6764	1292–21,674	
Female, *n* = 44	7449 (3972)	6585	1560–16,345	
Tumor size, cm				0.086
≤2, *n* = 49	6750 (3441)	5997	1560–15,937	
>2, *n* = 51	8131 (4448)	6888	1292–21,674	
Lymph node involvement				<0.001
Negative, *n* = 57	6173 (3087)	5492	1560–15,937	
Positive, *n* = 43	9154 (4509)	7569	1292–21,674	
Tumor cell differentiation				0.185
Well, *n* = 20	6385 (3207)	5601	2620–14,819	
Moderate and poor, *n* = 80	7722 (4181)	6778	1292–21,674	
Lymphovascular invasion				0.029
Negative, *n* = 65	6812 (3891)	5768	1560–21,674	
Positive, *n* = 35	8648 (4058)	7569	1292–16,345	
Perineural invasion				0.052
Negative, *n* = 65	6881 (3492)	5997	1560–16,345	
Positive, *n* = 35	8521 (4739)	7241	1292–21,674	
CA 19-9, U/mL				0.281
Normal ≤ 37, *n* = 45	7016 (3963)	6256	1743–21,674	
Abnormal > 37, *n* = 43	7945 (4065)	7084	1292–15,937	
CEA, ng/mL				<0.001
Normal ≤ 5, *n* = 74	6924 (3531)	6338	1292–15,937	
Abnormal > 5, *n* = 14	11,222 (5051)	11,627	4567–21,674	
Stage				0.028
I, *n* = 39	6351 (3201)	5697	1560–15,937	
II and III, *n* = 61	8160 (4355)	6893	1292–21,674	

SD: standard deviation; CA 19-9: carbohydrate antigen 19-9; CEA: carcinoembryonic antigen.

**Table 2 cancers-13-02313-t002:** Prognostic factors by univariate analysis for ampullary cancer patients undergoing pancreaticoduodenectomy.

Prognostic Factors	Survival Time, mon. Mean (SD)	Survival Time, mon. Median	Survival Time, mon. Range	1-Year Survival	5-Year Survival	*p*
Total, *n* = 100	52.3 (36.7)	44.5	0.2–181.0	92.0%	66.2%	
cfDNA level, copies/mL						<0.001
Low (≤6687), *n* = 50	69.2 (39.7)	59.5	15.6–181.0	100%	82.5%	
High (>6687), *n* = 50	35.4 (23.7)	34.3	1.6–103.8	84.0%	49.9%	
Tumor size						0.671
≤2, *n* = 49	54.9 (38.9)	45.1	1.4–181.0	89.8%	63.2%	
>2, *n* = 51	49.8 (34.6)	42.8	0.2–177.3	94.1%	69.4%	
Lymph node involvement						0.009
Negative, *n* = 57	62.4 (40.4)	55.7	4.5–181.0	93.0%	75.5%	
Positive, *n* = 43	39.0 (26.0)	37.8	0.2–112.8	90.7%	53.6%	
Tumor cell differentiation						0.248
Well, *n* = 20	62.6 (31.9)	62.1	7.0–128.0	90.0%	79.3%	
Moderate and poor, *n* = 80	49.8 (37.6)	40.0	0.2–181.0	92.5%	62.8%	
Lymphovascular involvement						0.879
Negative, *n* = 65	57.4 (41.1)	50.2	1.2–181.0	90.8%	66.0%	
Positive, *n* = 35	42.9 (24.5)	40.1	0.2–112.8	94.3%	66.5%	
Perineural invasion						0.748
Negative, *n* = 65	55.1 (37.7)	44.5	1.4–181.0	93.8%	67.4%	
Positive, *n* = 35	47.2 (34.7)	41.7	0.2–155.6	88.6%	64.0%	
CA 19-9, U/mL						0.071
Normal ≤ 37, *n* = 45	57.0 (37.6)	50.2	4.5–181.0	95.6%	77.9%	
Abnormal > 37, *n* = 43	46.2 (31.0)	41.8	1.4–141.4	88.4%	61.4%	
CEA, ng/mL						0.031
Normal ≤ 5, *n* = 74	54.8 (36.0)	47.3	0.2–181.0	91.9%	72.3%	
Abnormal > 5, *n* = 14	31.0 (22.1)	24.4	1.4–80.6	85.7%	50.0%	
Stage						0.002
I, *n* = 39	67.9 (42.6)	61.2	4.5–181.0	92.3%	83.7%	
II and III, *n* = 61	42.4 (28.5)	38.2	0.16–141.4	91.8%	54.7%	

SD: standard deviation; cfDNA: circulating cell-free deoxyribonucleic acid; CA 19-9: carbohydrate antigen 19-9; CEA: carcinoembryonic antigen.

**Table 3 cancers-13-02313-t003:** Multivariate analysis of the independent prognostic factors in ampullary cancer patients undergoing pancreaticoduodenectomy.

Prognostic Factor	Odds Ratio (95.0% CI)	*p*
cfDNA level, copies/mL	2.729 (1.225–6.079)	0.014
Low (≤6687), *n* = 50		
High (>6687), *n* = 50		
Lymph node involvement	1.051 (0.393–2.814)	0.920
Negative, *n* = 57		
Positive, *n* = 43		
CEA, ng/ml	1.395 (0.546–3.586)	0.487
Normal ≤ 5, *n* = 74		
Abnormal > 5, *n* = 14		
Stage	2.798 (1.056–7.412)	0.039
I, *n* = 39		
II and III, *n* = 61		

cfDNA: circulating cell-free deoxyribonucleic acid; CA 19-9: carbohydrate antigen 19-9; CEA: carcinoembryonic antigen.

## Data Availability

The data presented in this study are available on request from the corresponding author.

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
