# Peer review of "Circulating Cell-Free DNA as a Prognostic Biomarker in Resectable Ampullary Cancer"

_cancers, 2021, doi:10.3390/cancers13102313_

Round 1
Reviewer 1 Report
The article Circulating cell-free DNA in resectable ampullary cancer by Bor-Uei Shyr is interesting. Given the poor prognosis and the need for the early biomarker the study is the need of the hour.
The article elegantly laid out the reason for the study, results and discussion. If the authors adda para of why CEA and CA19.9 did not shon significance in their study, it would be a valuable resource for the readers.
As a reviewer I have few concerns:
- The major concern is the quantification method chosen for this study. qRT-PCR mostly give the relative transcript levels. Given the cell free DNA will be at avariosu lengths, the creation of standard curve will be an issue. Neither the methodology is not clear enough to describe the details not any reference to the method used.
- Second, the comparison with the controls. Given the age-related pathogenesis and the cell-free DNA could be from any other conditions the amount of cfDNA is always an issue. In this condition, the authors should reanalyze with age-matched controls and give the details in the revised version.
- The methodology for the CEA and CA19.9 should be given I the current manuscript. Otherwise, if this was part of another study, the authors should describe the details and obtain related permission to publish the data.
Author Response
Response to Reviewer 1 Comments
The followings are point-by-point response to the reviewer’s comments:
Point 1: The major concern is the quantification method chosen for this study. qRT-PCR mostly give the relative transcript levels. Given the cell free DNA will be at various lengths, the creation of standard curve will be an issue. Neither the methodology is not clear enough to describe the details not any reference to the method used.
Response 1: We have modified the quantification method chosen for this study in more details, “A TaqMan quantitative polymerase chain reaction (qPCR) assay (Thermo Fisher Scientific) of the housekeeping gene cyclophilin, which was not known to be correlated with cancer, was used to quantify the cfDNA copy numbers in the plasma samples. qPCR was performed using TaKaRa Ex Master Mix (Takara Bio, Shiga, Japan) according to the manufacturer's instructions. The sequence of cyclophilin primers were as below: forward ACATGGGTACTAAGCAACAAAATAAG and reverse CACAATTGGAACATCTTTGTTAAAC. The probe primer was Fam-TTGCAGACAAGGTCCCAAAGACAGCA-Tamra. Serially diluted standard DNA was used to generate a standard curve. The results were expressed as the threshold cycle (Ct), which was the cycle number at which the PCR product crossed the threshold of detection. To reduce the batch effect, we prepared a large volume tube of pre-mixed plasma samples (20 ml pooled from multiple samples) and prepared small aliquots of the pooled samples in standard tubes (1 ml) for storage at −80°C. When performing cfDNA extraction and qPCR experiment, we used plasma samples from clinical individuals and the pre-mix standard tube. The cfDNA copy number in each patient was measured according to the Ct value and the standard curve from serially-diluted DNA (0.001, 0.01, 0.1, 1, 10, 100 ng). The results of standard tubes between different batches were used to calculate the batch-effect factor for adjusting the copies/mL value in the following analyses. The batch-effect factor was calculated based on the pre-mix standard plasma cfDNA leve. Subsequently, the cfDNA copy number was normalized according to the plasma input volume and the batch-effect factor, and was expressed in copies/mL.” marked in red in the Materials and Mehtods section, from page 7 line 175 to 193.
Point 2: the comparison with the controls. Given the age-related pathogenesis and the cell-free DNA could be from any other conditions the amount of cfDNA is always an issue. In this condition, the authors should reanalyze with age-matched controls and give the details in the revised version.
Response 2: We have described “The cfDNA levels of 95 healthy volunteers without any history of malignancy were measured at our institute, including 66 males and 29 females with a mean age 54.2 ± 15.5 years. These healthy volunteers had no subsequent diagnosis of malignancy for at least two years after blood sampling.” marked in red in the Materials and Mehtods section, from page 7 line 193 to 196. We also have described “T The median of 6,687 copies/mL was used as the cut-off of 6,687 copies/mL for analysis. The levels of cfDNA in 95 normal healthy controls ranged from 0 to 4,157 copies/mL with a median of 168 copies/mL and a mean of 613 ± 888 copies/mL at our institute (Figure 1). The ROC curve (receiver operating characteristic curve) was shown in Figure 2.” marked in red in the Result section, from page 2 line 68 to 72.
Point 3: The methodology for the CEA and CA19.9 should be given in the current manuscript. Otherwise, if this was part of another study, the authors should describe the details and obtain related permission to publish the data.
Response 3: We have described the methodology for the CEA and CA19.9 by adding “Method for measuring CA 19-9 was radioimmunoassay and CEA was based on chemiluminescence in serum during enzyme-linked immunosorbent assays (ELISA).” marked in red in the Materials and Mehtods section, from page 7 line 165 to page 6 line 167.
Reviewer 2 Report
The manuscript “Circulating cell-free DNA in resectable ampullary cancer” by Bor-Uei Shyr et al addresses a relevant and innovative topic about the detection and analysis of circulating cell-free DNA as surrogate biomarker for diagnosis and prognosis in cancer.
There are several comments:
- The title of the manuscript is not appropriate for a research article, it is too generic. In the present form, it could be appropriate for a review article.
- The Table 1 shows that the values of circulating cfDNA levels are significantly higher in patients with lymph nodes involvement, lymphovascular invasion, perineural invasion, and CEA levels. These data are interesting but not sufficiently innovative. Although the liquid biopsy has been reported to be an innovative diagnostic and prognostic approaches, it is note that high levels of circulating cfDNA may be associated with the cancer aggressiveness and invasion. The author should enrich this table with more biological information such as the presence of specific gene mutations in cfDNA.
- The table 2 is confusing. It is not clear to what are referred to, the values of “Mean” and “Median”.
Author Response
Response to Reviewer 2 Comments
The followings are point-by-point response to the reviewer’s comments:
Point 1: The title of the manuscript is not appropriate for a research article, it is too generic. In the present form, it could be appropriate for a review article.
Response 1: We have modified the title to be “Circulating cell-free DNA as a biomarker for prognosis in resectable ampullary cancer“ marked in red in title page, page 1 line 2 to 3.
Point 2: The Table 1 shows that the values of circulating cfDNA levels are significantly higher in patients with lymph nodes involvement, lymphovascular invasion, perineural invasion, and CEA levels. These data are interesting but not sufficiently innovative. Although the liquid biopsy has been reported to be an innovative diagnostic and prognostic approaches, it is note that high levels of circulating cfDNA may be associated with the cancer aggressiveness and invasion. The author should enrich this table with more biological information such as the presence of specific gene mutations in cfDNA.
Response 2: The association of high levels of circulating cfDNA with the cancer aggressiveness and invasion as shown in Table 1 and the biological information have been described and discussed by “The results revealed that the cfDNA level was significantly higher in patients with positive lymph node involvement or stage II and III cancer than in those with negative involvement or stage I cancer. In other words, the cfDNA level tended to increase with advanced ampullary cancer. These associations might suggest that cfDNA levels could be a reflection of the tumor burden in patients with ampullary cancer. Moreover, higher levels of cfDNA were also noted in patients with positive lymphovascular invasion and abnormal serum CEA levels. These findings imply that the cfDNA level could be a biomarker of the biological behavior of ampullary cancer.” marked in red in Discussion section, page 6 line 136 to 143.
Point 2: The table 2 is confusing. It is not clear to what are referred to, the values of “Mean” and “Median”.
Response 3: We have added “ Survival time, mon.” for these items in Table 2
Reviewer 3 Report
This article evaluates the levels of circulating cell-free DNA (cfDNA) as a biomarker in resectable ampullary cancer patients. The authors evaluated the levels of cfDNA in 100 ampullary cancer patients and in 95 normal healthy controls and found that cfDNA levels were elevated in cancer patients being associated with poor prognosis. These findings are interesting considering the implications of cfDNA in other tumor types.
I think this paper add important initial knowledge about the importance of cfDNA as a surrogate marker of ampullary cancer aggressiveness with huge potential to be used in the clinics. Overall the article is well written and well-illustrated.
However, some points should be addressed:
1 – The authors have evaluated cfDNA levels in 100 cancer patients and in 95 normal healthy controls, being significantly higher in cancer patients when compared with controls. Although the authors have described the median and mean for each group, I will ask to the authors to provide a graph/image with the cfDNA levels for all samples analyzed (controls vs cancer patients).
2 – The authors used the cut-off of 6,687 copies/mL for their analysis. I would like to ask why they decided to use this cut-off instead of using 4,157 copies/mL (highest number of copies found in the controls)? What would be the results if 4,157 cut-off was used? (line 66, please change 4157 for 4,157 to standardize all numbers).
3- In alternative the authors can also show ROC curves for discriminate patients with cancer from healthy volunteers.
4- It has been described that DNA released from dead cancer cells varies in size, while DNA released from apoptotic “healthy” cells are more uniform in size. I would like to ask if the authors have tested DNA integrity in their samples. If not, could they perform this experiment to validate the integrity/fragment size of their samples.
5 – In line 84-86 the authors descrived the results regarding 1- and 5-year survival rates and mention Figure 1. However, it is hard to visualize the 1- and 5-year survival in the graph provided (especially the 1-year survival). I will suggest to indicate within the kaplan-Meier curves the intersection of 1- and 5-year survival.
6 – In the discussion section the authors should discuss the applicability of cfDNA in other tumor types in order to show the great importance of their findings.
7 – Material and Methods section should be divided into sub-sections: such as Samples; cfDNA quantification, statistical analysis….
8 - I would like to ask to provide the information regarding the primers were used in this work (which Alu sequences?). Moreover it will be of importance to provide the sensitivity/efficiency of ALU-qPCR used (using purified genomic DNA and cfDNA).
I have no other real comments.
Author Response
Response to Reviewer 3 Comments
The followings are point-by-point response to the reviewer’s comments:
1 – The authors have evaluated cfDNA levels in 100 cancer patients and in 95 normal healthy controls, being significantly higher in cancer patients when compared with controls. Although the authors have described the median and mean for each group, I will ask to the authors to provide a graph/image with the cfDNA levels for all samples analyzed (controls vs cancer patients).
Response 1: We have added “Figure 1. The scatterplot for ampullary cancer (study group) and healthy volunteers (normal control)”
2 – The authors used the cut-off of 6,687 copies/mL for their analysis. I would like to ask why they decided to use this cut-off instead of using 4,157 copies/mL (highest number of copies found in the controls)? What would be the results if 4,157 cut-off was used? (line 66, please change 4157 for 4,157 to standardize all numbers).
Response 2: We has added “The median of 6,687 copies/mL was used as the cut-off of 6,687 copies/mL for analysis.” marked in red in Result section, page 2 line 68 to 69. The reason to use median of study group is to equally divide the study group patients, and to avoid small sample size in one side for analysis. We also change 4157 for 4,157 to standardize all numbers, marked in red in Result section, page 2 line 70.
3- In alternative the authors can also show ROC curves for discriminate patients with cancer from healthy volunteers.
Response 3:
4- It has been described that DNA released from dead cancer cells varies in size, while DNA released from apoptotic “healthy” cells are more uniform in size. I would like to ask if the authors have tested DNA integrity in their samples. If not, could they perform this experiment to validate the integrity/fragment size of their samples.
Response 4: We did not test DNA integrity in our samples. Thank you for reminding, we would like to do so if we have enough samples in the future.
5 – In line 84-86 the authors described the results regarding 1- and 5-year survival rates and mention Figure 1. However, it is hard to visualize the 1- and 5-year survival in the graph provided (especially the 1-year survival). I will suggest to indicate within the kaplan-Meier curves the intersection of 1- and 5-year survival.
Response 5: “Figure 1” has been modified with “indication of 1- and 5-year survivals” within the Kaplan-Meier curves.
6 – In the discussion section the authors should discuss the applicability of cfDNA in other tumor types in order to show the great importance of their findings.
Response 6: We have added “The applicability of cfDNA in other tumor types has been reported such as pancreatic cancer and stomach cancer [1, 3, 6, 9].”, marked in red in Discussion section, page 6 line 134 and line 135.
7 – Material and Methods section should be divided into sub-sections: such as Samples; cfDNA quantification, statistical analysis….
Response 7: We have added sub-sections, including cfDNA quantification, Statistical analysis, marked in red in Material and Methods section, page 7 line 168 and line 196.
8 - I would like to ask to provide the information regarding the primers were used in this work (which Alu sequences?). Moreover it will be of importance to provide the sensitivity/efficiency of ALU-qPCR used (using purified genomic DNA and cfDNA).
Response 8: We have provided the information regarding the primers by adding “The sequence of cyclophilin primers were as below: forward ACATGGGTACTAAGCAACAAAATAAG and reverse CACAATTGGAACATCTTTGTTAAAC. The probe primer was Fam-TTGCAGACAAGGTCCCAAAGACAGCA-Tamra. Marked in red in Material and Methods section, page 7 line 179 to 181.
Round 2
Reviewer 1 Report
The revised version address the comments raised in the previous review
Author Response
Thank you for reviewing.
Reviewer 2 Report
The authors well addressed previous comments.
Author Response
Thank you for reviewing and comments.
Reviewer 3 Report
The authors have addressed all the comments of the reviewers and revised the manuscript accordingly.
Author Response
Thank you for reviewing and comments.